# Hepatitis E Virus in the Role of an Emerging Food-Borne Pathogen

**DOI:** 10.3390/microorganisms13040885

**Published:** 2025-04-12

**Authors:** Alica Pavlova, Bozena Kocikova, Michaela Urda Dolinska, Anna Jackova

**Affiliations:** Department of Epizootiology, Parasitology and Protection of One Health, University of Veterinary Medicine and Pharmacy in Kosice, Komenskeho 73, 041 81 Kosice, Slovakia; alica.pavlova@student.uvlf.sk (A.P.); bozena.kocikova@uvlf.sk (B.K.); michaela.urda.dolinska@uvlf.sk (M.U.D.)

**Keywords:** hepatitis E virus, HEV infection, food-borne transmission, food-producing animals, animal products

## Abstract

Viral hepatitis E represents an important global health problem caused by the hepatitis E virus (HEV). Cases of HEV infection are increasingly associated with food-borne transmissions after the consumption of raw or undercooked food products from infected animals in high-income regions. Although most cases of infection are asymptomatic, severe courses of infection have been reported in specific groups of people, predominantly among pregnant women and immunocompromised patients. The viral nucleic acid of HEV is increasingly being reported in food-producing animals and different products of an animal origin. Even though the incubation period for HEV infection is long, several direct epidemiological links between human cases and the consumption of HEV-contaminated meat and meat products have been described. In this article, we review the current knowledge on human HEV infections, HEV in different food-producing animals and products of an animal origin, as well as the accumulation and resistance to HEV in farm and slaughterhouse environments. We also provide preventive measures to help eliminate HEV from animals, the human population, and the environment.

## 1. Introduction

The hepatitis E virus (HEV) is a small virus, ranging from 27 to 34 nm in diameter, with its genome composed of a single-stranded, positive-sense RNA [1]. Two different infectious HEV particles exist, namely the non-enveloped particles shed in the faeces and the quasi-enveloped particles found in the bloodstream [2]. The quasi-enveloped particles are cloaked in host cell membranes that are thought to protect the virus from neutralisation by the host antibodies [3].

HEV belongs to the family *Hepeviridae* within the genus *Paslahepevirus*. Eight genotypes have been described in the species *Paslahepevirus balayani*, five of which are significant causative agents of human diseases [4]. Genotypes 1 and 2 (HEV-1 and HEV-2) infect only humans, whereas HEV-3 and HEV-4 are zoonotic and are naturally present in several animal species. One zoonotic case was also described with HEV-7 after the consumption of camel meat and milk [5]. Until now, genotypes HEV-5, HEV-6, and HEV-8 have only been detected in animal populations [6]. The majority of human viral hepatitis E cases in Europe are linked to the zoonotic genotype HEV-3 [7]. This genotype’s main transmission route is through the consumption of raw or undercooked meat and meat products [8].

The presence of HEV has been genetically detected in various naturally infected domestic and wild animals [9]. HEV infections in animals result in a subclinical course of infection, with only mild microscopic lesions [10]. The presence of HEV RNA has also been repeatedly reported in different products of an animal origin [11,12,13].

Infection in the human population can be highly underdiagnosed, since it is usually asymptomatic in immunocompetent patients [2]. However, the clinical manifestation of infection depends on the HEV genotypes and the patient’s immune system. Two population groups are particularly susceptible to developing symptomatic HEV infections, including pregnant women and immunocompromised patients [14,15].

As most cases in Europe are locally acquired infections caused by the zoonotic genotype HEV-3, it is crucial to identify the possible sources of HEV infection and prevent food-borne outbreaks. The aim of the present study is to summarise the current knowledge about the occurrence of HEV in different food-producing animals and their products that represent possible risks of food-borne transmissions for consumers.

## 2. Epidemiology of HEV

Previously, HEV was thought to be an infection limited to several low-income regions in Asia, Africa, the Middle East, and Mexico, along with travel-associated infection in high-income regions in Asia, Europe, and North America [2]. Currently, it is known that HEV is an important infectious agent worldwide. Epidemics of viral hepatitis E are typical for low-income regions, especially during rainy seasons. The first recorded outbreak was in India between 1955 and 1956, with more than 29,000 symptomatic cases [16]. Numerous outbreaks have also been described in China, Bangladesh, Nepal, and Uganda [17,18,19,20]. However, sporadic autochthonous HEV infections have been reported in many high-income regions [2,7].

The true prevalence of HEV among humans is unknown due to frequent asymptomatic or unrecognised infections. The number of subclinical human infections is at least two times greater than clinical infections among sporadic cases and during outbreaks in low-income regions [19]. Clinical symptoms of HEV infection can only be seen in a small population of adults (1–2%) [21]. The worldwide HEV seroprevalence in the human population varies depending on the population group being tested or the geographical region. Every year, approximately 20 million people are infected with HEV, leading to over three million clinical cases and 70,000 deaths [22]. A recent meta-analysis reported the global HEV prevalence by retrieving data from 75 countries. It was found that more than 900 million individuals have experienced past HEV infections based on their seropositivity for anti-HEV antibodies. A high seroprevalence has been described in Africa (21.76%) and Asia (15.80%), with a lower prevalence in Europe (9.31%), North America (8.05%), and South America (7.28%) [23]. A positive correlation exists between age and HEV seropositivity, where the highest seropositivity rates are observed in adults over sixty years of age [24]. Over recent years in Europe, the number of confirmed cases has risen continuously, with a tenfold surge from 514 cases in 2005 to 5617 in 2015 [25]. Data from more seroprevalence studies has been published in Europe, and some specific areas in the United Kingdom, France, and Germany are considered to be endemic [26,27,28]. There is limited information regarding the variation in the awareness and testing in many European countries. Moreover, HEV is not subject to EU-wide surveillance, which further leads to difficulties in determining the prevalence of HEV infections in Europe [29].

## 3. Transmission of HEV

HEV genotypes that cause viral hepatitis E in the human population can be transmitted in many ways, depending on the region and the HEV genotype. In low-income regions, the main transmission route is water-borne infections caused by genotypes HEV-1 and HEV-2 [22,30]. Until now, the vertical transmission of HEV from a mother to the foetus during pregnancy and blood-borne infections have also been described [31,32].

In high-income regions, most infections are caused by genotypes HEV-3 and HEV-4 being transmitted through animal sources. A high identity has been often described between the nucleotide sequences of the HEV strains obtained from humans and animals [33,34]. Zoonotic transmission occurs via direct contact with infected animals and environmental contamination by animal faeces in an agricultural sphere [35]. Food-borne transmissions through raw or undercooked products of an animal origin are considered the most important ways of transmitting these genotypes [34,36]. Sporadic cases of infection have also been linked to blood-borne infections and the transplanted organs of infected humans [37,38]. Infections caused by HEV-7 have been reported in patients after the consumption of camel meat and milk [5]. Figure 1 displays the different HEV transmission routes in the human population.

### 3.1. Food-Borne Transmission of HEV

Most cases of HEV infection in high-income regions have been linked to the ingestion of raw or undercooked meat, liver, and liver sausages from infected domestic or wild animals. The long incubation period creates difficulties in proving a causal link, because the food has often been eaten or discarded before the clinical symptoms of infection appear. However, there are several direct epidemiological links between human cases and the consumption of HEV-contaminated meat and meat products.

Four patients with HEV infections have been reported as consuming raw meat from a wild sika deer in Japan. In three cases, there was a 100% nucleotide identity, and in one case, a 99.7% identity with the consumed meat [36].Consumption of grilled wild boar meat has been confirmed as a source of infection in patients from Japan. The similarity between the patient and the meat sequences was 99.95% [39].A mini outbreak of HEV infection in family members has also been described in Japan. The source of the infection was grilled pork meat [40].Two cases of infection caused by the genotype HEV-4 have been described in France. Patients confirmed the consumption of uncooked pork liver sausages. The genetic sequences were 96.7% identical to the European swine sequences [33].A direct HEV transmission through ingesting raw pig liver sausage figatelli has been described in France. Sequences from the case of HEV infection and the consumed sausages were 100% identical [41].A patient described an acute infection of HEV after consuming pork meat from a local family farm in Spain. The same strain of the genotype HEV-3 was sequenced from meat consumed by the patient [42].A locally acquired outbreak of HEV infection caused by the genotype HEV-3 has been described in Australia [43].Chronic infections of HEV-7 have been reported in patients after the consumption of camel meat and milk [5].A family outbreak has been described in Spain. Eight members of a family were all infected with HEV-3 after consuming wild boar meat. The HEV sequences from a patient were 100% identical to the wild boar meat [34].

Although there are only few direct epidemiological links between cases, HEV is considered to be an emerging food-borne pathogen. The data concerning HEV in animal populations are expanding annually. Numerous studies have reported the circulation of HEV in different food-producing animals and products of animal origin worldwide (Appendix A, Appendix A).

#### 3.1.1. Presence of HEV in Food-Producing Animals

Zoonotic HEV was first discovered in 1997 among domestic pigs in the United States [44]. Pigs have been considered as the most important reservoirs of HEV. Anti-HEV antibodies and HEV RNA have been detected in pigs from the countries of all continents except Antarctica. The circulation of HEV has been confirmed in pig’s sera or faeces from Africa [45,46,47,48], Asia [49,50,51,52,53], North America [54,55,56], South America [57,58,59,60,61], and Australia [62]. Anti-HEV antibodies have been reported in the pig farms of many European countries [63,64,65,66,67]. Similarly, the presence of HEV RNA has been described in pigs [68,69,70,71,72,73]. HEV shedding occurs not only in faeces but also in urine, which may contribute to the spread of the virus in the environment and pose the risk of HEV transmission [74,75].

Several studies have also detected anti-HEV antibodies and HEV RNA in domesticated ruminants [55,76,77,78,79,80]. Even though HEV has been confirmed in the cow population, a direct detection of viral RNA in Europe was unsuccessful [70,81,82]. The susceptibility of sheep and goats to HEV infection has been assessed in more molecular surveys. The detection of viral RNA has been reported in Italy, Mongolia, and China [83,84,85,86]. Cross-species transmissions between ruminants and pigs are possible due to the detection of genetically closely related HEV strains in the same geographical areas [83]. The relatively low prevalence rates in domesticated ruminants suggest that they are not a reservoir of HEV. Infections of HEV can also be caused by the spillover events [83]. However, more studies are needed to evaluate the presence of HEV in the ruminant population.

HEV circulation has also been described in wildlife species, mainly in wild boar and deer populations [36,87]. Wild boars are considered to be another vital reservoir of HEV, which has been confirmed by an increasing number of studies in Europe [88,89,90,91,92]. Anti-HEV antibodies and HEV RNA have been found in red deer, roe deer, and fallow deer [78,93,94]. Identical HEV sequences in wild boar, deer, and other animal species in the same geographical areas are common [8,95]. This is probably caused by interspecific interactions in the same area, especially those with a high animal density. Aggregations around feeding sites often occur during the year when feeding supplies are reduced [70].

Rabbits are natural hosts of the rabbit HEV strain (HEV-3ra), which is classified within HEV-3. The first detection of HEV-3ra was published in China [96]. The presence of HEV-3ra has also been described in the United States, Egypt, Japan, The Netherlands, France, and Germany [97,98,99,100,101,102]. Studies from Italy and Portugal have confirmed anti-HEV antibodies, but none of the samples were positive for HEV RNA [103,104]. Currently, the awareness of the possible transmission of this strain has risen, because human cases of viral hepatitis E have been connected with HEV-3ra in France, Ireland, Switzerland, and Germany [105,106,107,108].

#### 3.1.2. Presence of HEV in Meat and Offal

Animal by-products (offal) include all parts of an animal that are not part of the dressed carcass, such as the liver, heart, rumen contents, kidney, blood, fats, and spleen, and are suitable for human consumption [109]. Since HEV primarily replicates in hepatocytes, the liver can be a significant source of infection, especially in reservoir animal species [110]. In contrast to many other food-borne viruses, the viral contamination of HEV in meat products is not restricted to the food surface and is localised inside many different organs [75]. The detection of HEV RNA has also been described in the muscles, blood, kidney, spleen, and the heart [12,94,111,112]. A wide range of animals are susceptible to infections without clinical disease symptoms. Therefore, visual inspections cannot detect infected animals during slaughter [113].

The human risk of food-borne HEV infection depends on the infection status of the slaughtered animals [35]. Several risk factors seem important, such as the slaughter age, genetic background, lack of hygiene measures, and drinking water origins [114]. The immune status of animals is another critical factor. Co-infections of pigs with HEV and the porcine reproductive and respiratory syndrome virus (PRRSV) increase the possible risk, and HEV-containing livers can be present at the slaughter time [115].

Since pigs are considered to be the most essential HEV reservoirs, viral RNA can be detected throughout the complete pork food chain [1]. Generally, HEV RNA is more prevalent in pork liver and blood than in pork muscle [12,111,116]. HEV positivity in pork liver tissue has been described in numerous countries worldwide, including in North and South America, Asia, and Africa [60,61,117,118,119,120,121]. The prevalence of HEV RNA varies considerably, especially across many European countries (0.3–30.8%). Pig livers have been described as being positive in Switzerland (0.3%), Poland (1.0%), Spain (3%), the United Kingdom (3%), Germany (4%), France (4%), Czech Republic (5%), Italy (6%), The Netherlands (6.5%), Ireland (24%), and Hungary (30.8%) [12,81,122,123,124,125,126,127,128]. Viral RNA was also described in the blood and serum samples of slaughtered animals, being highly prevalent in blood and blood-derived products such as haemoglobin, fibrinogen, and plasma [111,112,129].

A direct detection of HEV RNA in the liver tissue of domesticated ruminants is relatively rare. It has been reported in bovine livers from Korea and Brazil [130,131]. Few studies have been published in Europe, but no positivity of HEV RNA has been detected in liver and blood samples [70,81,82]. The presence of viral RNA was confirmed in sheep livers from China and Mongolia, but there is no evidence from sheep and goat livers in Europe [86,132].

Another important category of animal products is from non-domesticated animals. Wild boar, red deer, roe deer, and fallow deer are usually hunted and slaughtered, mostly for private consumption, but their meat and meat products can also be found in specific markets or restaurants. Numerous studies have confirmed the widespread circulation of HEV in meat and offal from wild boars. Genotype HEV-3 is commonly isolated from wild boars in many European countries (1.9–33.5%) [65,70,81,94,133,134,135,136]. HEV RNA was found in the livers, blood, spleens, kidneys, and muscles. Viral RNA has also been detected in the meat and offal from red deer, roe deer, and fallow deer populations. Several studies have detected the virus in deer livers, with the prevalence of HEV RNA varying between 1.7 and 34.4% [70,81,94,133,137]. A study from Germany has confirmed that HEV RNA could also be present in deer kidneys, spleens, and muscles [94].

The presence of HEV-3ra has been described in the liver tissue of wild rabbits in The Netherlands, France, and Germany (17.1–60%) [98,99,100].

#### 3.1.3. Presence of HEV in Meat Products

The viral nucleic acid of HEV has been repeatedly detected in numerous meat products, especially in pork liver foods. After the consumption of raw pig liver sausages, which were directly linked to human cases, it was confirmed that meat products could be an important source of infection [41]. More studies have identified that eating raw or undercooked meat products is a higher risk factor for HEV infection. Still, the results can be influenced by the study methodology, geographical location, and local cuisines. The consumption of pork liver sausages and offal was found to be a significant risk factor in France [138,139]. A study in the United Kingdom described that ready-to-eat pork products may be an essential transmission source [140]. In Germany, uncooked wild boar meat and the consumption of offal were significantly associated with HEV infections [141].

The concentration of HEV in animal livers can be high, so a few livers from viraemic animals can contaminate a whole batch of sausages [142]. The risk of infection can also be partially influenced by the ingredients of meat products, traditional food processing, and different consumption habits occurring in different regions. In some countries, such as France or Germany, liver is an essential ingredient in the processing of sausages. The prevalence among the resulting products was found to range from 17 to 58% [142,143,144]. Food processing technologies, such as smoking, drying, or curing, can affect the infectivity of HEV. Another critical point can be small traditional homemade food processing practices that can raise the possible risk of infection [145]. Consumption habits are also important. In some countries, specific meat products are typically eaten raw or briefly cooked [12].

The occurrence of HEV RNA has been confirmed in meat products from South Africa, Canada, and Brazil [120,146,147,148], whereas no samples have tested positive in Argentina [149]. In contrast, HEV RNA has been described in raw liver sausages, raw pork sausages, salami, raw dried ham, liver pâté, raw wild boar sausages, and wild boar homemade liver sausages in many European countries. Two studies from The Netherlands have detected the nucleic acid of HEV in their typical meat products, including liver pâtés, liverwurst, cervelaat, salami, metworst, and snijworst [150,151]. HEV RNA was also found in liver sausages and liver quenelles in France, as well as in raw pork and wild boar sausages in Germany, wild boar sausages, raw and dry liver sausages in Italy, liver pâtés and raw dry hams in Belgium, pork sausages in Ireland, and liver and raw meat sausages in Switzerland [128,142,144,145,152,153].

#### 3.1.4. Presence of HEV in Milk

Recently, awareness of the possible milk-borne transmission of HEV has risen. Even though the knowledge of HEV within milk is relatively limited, the role of mammary glands as a source of HEV RNA excretion has been investigated, and it was found that HEV can be present in breast milk during the acute phase of HEV infection [154]. The risk of the milk-borne transmission of HEV has also been confirmed by the fact that primates (rhesus macaques) can be experimentally infected by milk from HEV-infected cows [77]. Possible milk-borne infection has been described after camel milk consumption in the United Arab Emirates [5].

A growing number of reports demonstrate the presence of HEV RNA in the milk of domesticated ruminants, such as cows, sheep, and goats. However, the occurrence of HEV in animal milk is variable, and a higher prevalence of HEV RNA can be visible in regions of Asia and Africa while being lower in Europe and America [155]. This is probably connected to the level of hygiene standards and breeding techniques. The regions with traditional small mixed farms with more animal species (pigs, cows, goats, and sheep) can be at a higher risk of cross-species infections [77,84,155].

Up to now, the knowledge of viral RNA within milk has been limited. Only a few studies about HEV RNA in the milk of ruminants have been published, and the presence of viral RNA in cow milk has been confirmed in Egypt (0.2%), Turkey (29.2%), and China (37%) [77,156,157]. In these studies, different genotypes have been detected as follows: in Egypt, HEV-3; in China, HEV-4; and surprisingly in Turkey, the genotypes HEV-1, HEV-3, and HEV-4. The detection of HEV-1 in animal milk is an unusual finding. Further studies are required to confirm the possible presence of this genotype in animal samples. Cow milk was also screened for HEV RNA in Germany and Belgium, but the samples were all found to be negative [155,158]. HEV positivity has been observed in milk samples from small ruminants in Egypt, Turkey, the Czech Republic, and Italy. In a study from Egypt, 0.7% samples of goat milk were positive for the genotype HEV-3 [159]. In Turkey, 12.3% of ovine milk samples and 18.46% of goat milk samples were found to be positive [156]. In the Czech Republic, 2.8% of mixed ovine and goat milk samples were positive [13], while in Italy, positivity in ovine milk was 2.27% with the detection of HEV-3 [160]. As viral particles of HEV can be present in milk, the production of traditional milk products from raw milk in specific European regions represents a significant public health concern.

#### 3.1.5. Presence of HEV in Shellfish

A marine environment contaminated by HEV can lead to the accumulation of the virus in the digestive tissues of shellfish. This poses a risk of the shellfish-borne transmission of this virus in the human population [11]. Shellfish represent an important HEV transmission source, since during their filter-feeding process, they can concentrate HEV viral particles from the surrounding marine environment. Therefore, they can be an ideal natural indicator of the presence of HEV in environmental waters [161]. As most shellfish are usually eaten raw, viable virus particles can pose a risk to public health. Although relatively rare, the consumption of raw or undercooked shellfish has been identified as a possible source of HEV infection. In 2004, the presence of viral hepatitis E in some Japanese patients was connected with shellfish consumption [162]. Later, an outbreak of HEV was confirmed in passengers returning from a world cruise trip, where the potential source of the infection was considered to be of a shellfish origin [163].

Prevalence studies of HEV in shellfish have been reported in more European countries. Whilst a few studies point to the absence of HEV in shellfish [164,165,166,167], other studies have documented a variable HEV presence. A high prevalence of HEV RNA was observed in the United Kingdom, where 85.4% of mussels were found to be positive for HEV RNA. A phylogenetic analysis showed that most sequences were clustered with the genotype HEV-3 from humans and swine [11]. The occurrence of HEV-3 in different kinds of shellfish was also reported in Spain, Italy, and Scotland [168,169,170].

Differences found in the HEV RNA prevalence in shellfish seem to be influenced by the following factors: type of the examined shellfish, season of the sampling, place of the sampling, contamination of the water environment, seasonality, and age of the shellfish (with a longer period, a higher accumulation of viral particles can be found) [161,166,171].

## 4. Clinical Manifestations of HEV Infection in the Human Population

### 4.1. Acute HEV Infection

An HEV infection is usually a self-limiting asymptomatic infection in the majority of patients. The incubation period of the HEV infection ranges from 2 to 8 weeks, with an average of 5 to 6 weeks [172]. During the acute phase of the infection, hepatic and extra-hepatic manifestations have been described [173]. Typical symptoms of a prodromal phase of an acute infection are malaise, fever, nausea, vomiting, pruritus, and a subsequent icteric phase, which includes dark urine and jaundice. In immunocompetent patients, the viremia can be cleared without treatment [174].

Clinical manifestations are highly variable in specific groups of patients, depending on the HEV genotypes and host immune system. Genotypes HEV-1 and HEV-2 are predominant in low-income countries in Asia or Africa (Figure 2), where the incidence of symptomatic infections is higher [22,175]. The hepatic manifestations following HEV-1 and HEV-2 infections vary from an entirely asymptomatic disease course or a mild systemic infection to icteric acute hepatitis and fulminant liver failure. A severe course of infection caused by these genotypes is visible in pregnant women. During pregnancy, the mortality rate may reach 30%, mainly due to obstetric complications such as haemorrhages or eclampsia [176]. Fulminant liver failure can also occur, with a ten-fold higher risk compared to non-pregnant patients [177]. Vertical transmission to infants also leads to increased neonatal morbidity and mortality [178].

The genotype HEV-3 is predominant in Europe and the United States (high-income regions), while the genotype HEV-4 has been documented primarily in Asia. Both genotypes have lower rates of symptomatic presentation [14,179,180]. However, specific patient groups seem to carry a higher risk for HEV infection, especially with regard to HEV-3 [7]. In Europe, predominantly immunocompromised patients suffer from severe symptoms of infection. The main contributing factors have been identified as the male sex, age above 50 years, pre-existing liver disease, immunocompromised status, diabetes mellitus, and alcohol consumption [181,182,183]. In these patients, acute hepatitis, fulminant liver failure, and also chronic HEV infections can be observed.

In some parts of world, most often in middle-income regions, cocirculation between genotypes can be seen. In South America, the predominant genotype is HEV-3, but cases of HEV infection caused by HEV-1 have also been described [184]. Similarly, in some regions of Asia, HEV-4 in cocirculation with HEV-1 can be seen [185] (Figure 2).

Extra-hepatic manifestations of HEV infection are poorly understood. However, its occurrence is probably caused by the host’s systemic immune response in extra-hepatic organs and its viral replication in non-hepatic tissues [186]. Neurological disorders have been the most widely documented, especially Guillain–Barré syndrome, neuralgic amyotrophy, encephalitis, meningoencephalitis, and myositis [187,188]. In France, 16.5% of patients infected with HEV reported neurologic symptoms [189]. Guillain–Barré syndrome associated with an acute HEV infection has been described in 5% of patients in The Netherlands [190]. Acute hepatitis E has also been found in 10% of patients with neuralgic amyotrophy in the United Kingdom and The Netherlands [191]. In more studies, patients with HEV-associated neurological manifestations generally had a normal liver function, and the neurological symptoms were dominant in these patients. Thus, there is a possibility that HEV is directly neurotropic and can possibly be replicated in the nervous system [188]. In addition to neurological diseases, various extra-hepatic manifestations such as renal disorders, haematologic complications, pancreatitis, thyroiditis, myositis, and myocarditis have been associated with HEV [173,192,193,194].

### 4.2. Chronic HEV Infection

Chronic HEV infections have become a serious problem in immunocompromised patients, resulting in substantial morbidity and even mortality [195]. The first cases of chronic HEV infections were reported in organ transplant recipients in 2008 [196]. According to this study, patients who are viremic for more than 3 months can be regarded as being chronically infected. Cases of chronic HEV infections have also been described in patients infected with the human immunodeficiency virus (HIV), patients with autoimmune diseases, and oncological patients receiving chemotherapy and immunotherapy [197,198,199]. The genotype HEV-3 was the cause in all of these cases. Chronic infection with the genotype HEV-4 was first reported in patients with leukaemia during chemotherapy treatments in China [200]. Until now, chronic HEV infections caused by HEV-1 and HEV-2 have not been observed [201].

Although in most patients with chronic HEV infections, no symptoms and only a mild elevation of the liver enzymes were described, a severe course of infection can be seen in some patients. The most common symptoms are fatigue, diarrhoea, abdominal pain, arthralgia, and rarely, jaundice [15]. Even more serious are irreversible changes in the liver tissue, including nodules, fibrotic remodelling, cirrhosis, and death due to decompensated cirrhosis [202]. Approximately 10% of patients can develop cirrhosis within a few years of a chronic HEV infection [196].

## 5. Preventive Measures

HEV usually circulates in the environment as a non-enveloped virus, but it can also survive under extreme conditions [203]. The accumulation of viruses in the environment plays a pivotal role in the spread of infection within animal and human populations. The important places where HEV accumulations occur are farms and slaughterhouses, where HEV can persist in the farm environment for an extended period of time. HEV-3 shows a high level of stability on different surfaces, such as wooden, metal, and plastic objects. The infectious virus can be detectable on most surfaces for up to 8 weeks [204]. Infected animals, along with their meat and offal, can contaminate the entire slaughter lines with HEV. Since HEV has been confirmed on different slaughter line surfaces, it can also be found on workers’ hands [72,205]. Surface and hand disinfection are the main components of infection control. Recent studies have confirmed that HEV on surfaces cannot be efficiently inactivated with low concentrations of alcohol-based disinfectants. Therefore, more effective disinfectants based on quaternary ammonium compounds, oxygen radicals, and peracetic acid are recommended to be utilised [204,206]. High-level hygiene standards and the use of effective disinfectants are thus essential preventive measures in all areas of farms and slaughterhouses.

There is limited information about HEV survival in meat, meat products, and shellfish. The infectivity of HEV in food products and its transmission to consumers is not entirely understood. Recent data indicate that HEV can resist food processing techniques [207]. It has been confirmed that a heating process to an internal temperature of 71 °C for 20 min is necessary to inactivate HEV entirely [208]. Experiments with incubating contaminated pig liver homogenates at 56 °C for 1 h did not inactivate the virus. This temperature is equivalent to medium-to-rare cooking conditions in a restaurant, indicating that the consumption of undercooked or raw pig meats could pose a risk of food-borne HEV infections [116].

Similarly, milk heat treatment presents a vital step in preventing food-borne infection. Pasteurisation is standard process in dairy industries; however, in some cases, it is not effective. On the contrary, a milk boiling process with 100 °C for 3 min provides a complete sterilisation [77]. Consequently, there is a possible risk of food-borne infections, which can be connected with consuming unpasteurised milk and milk products, such as cheeses [209]. However, to prevent transmissions by different products of an animal origin, appropriate heat treatment of the food is still required in order to inactivate the virus.

One of the best preventive measures is the existence of an effective vaccine. Up to now, only one vaccine is available. Hecolin is registered and licenced for use in China [210]. The vaccine is administered in three doses with a 0-, 1-, and 6-month schedule. It provides 100% protection against the genotype HEV-1 during the first 12 months. It is also effective against the genotype HEV-4 [210,211,212]. Unfortunately, there is no information about its cross-protection against HEV-3.

Over the past 20 years, numerous essential studies have been published that have changed the understanding of HEV. However, there are still many knowledge gaps. Since there is a lack of surveillance for HEV in humans, animals, and animal products, the true HEV prevalence is unknown. More information is needed about the possible ways of transmission in human and animal populations, HEV survival in the environment, and the development of clinical infections (infective doses, host susceptibility, dose–response relationships, and the virulence of HEV strains). A better understanding of HEV biology, transmission, replication, and pathophysiology could help eliminate the virus in the environment, animal, and human population. Therefore, cooperation between the general population, scientists, human doctors, and veterinary doctors could significantly prevent HEV food-borne outbreaks and protect consumers.

## 6. Conclusions

The hepatitis E virus is an emerging food-borne pathogen. The presence of HEV RNA has been repeatedly described in food-producing animals, meat, offal, meat products, milk, and shellfish. The number of reported autochthonous sporadic HEV infections in Europe has increased, but standardised methods and the surveillance of HEV infections are still missing. Several direct epidemiological links between human cases and the consumption of HEV-contaminated meat and meat products have been described. Although HEV mainly results in an acute self-limiting infection, severe symptoms can be seen in specific population groups. Preventive actions must be taken to successfully inactivate HEV, prevent food-borne outbreaks, and protect vulnerable consumer populations to ensure the health of humans and animals, along with food and environmental safety.

## Figures and Tables

**Figure 1 microorganisms-13-00885-f001:**
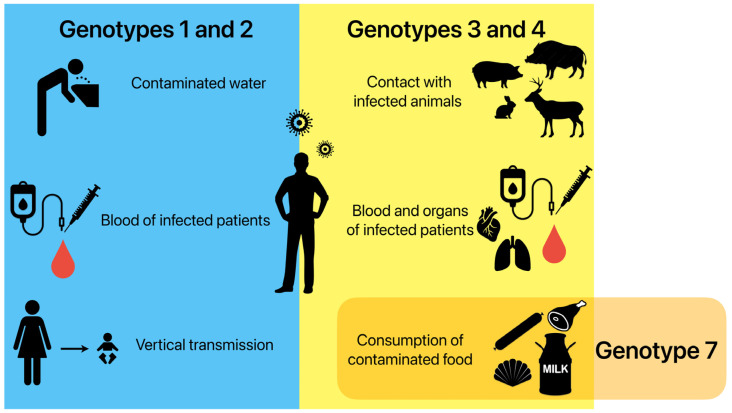
Possible ways of HEV transmission in the human population depending on genotypes. Genotypes 1 and 2 are mainly transmitted by drinking contaminated water, as well as transfusions of infected blood and from the mother to the foetus during pregnancy. Genotypes 3 and 4 are mainly transmitted by direct contact with infected animals, transfusions of infected blood and transplantations of infected organs, and the consumption of contaminated food. Genotype 7 is connected to the consumption of contaminated food.

**Figure 2 microorganisms-13-00885-f002:**
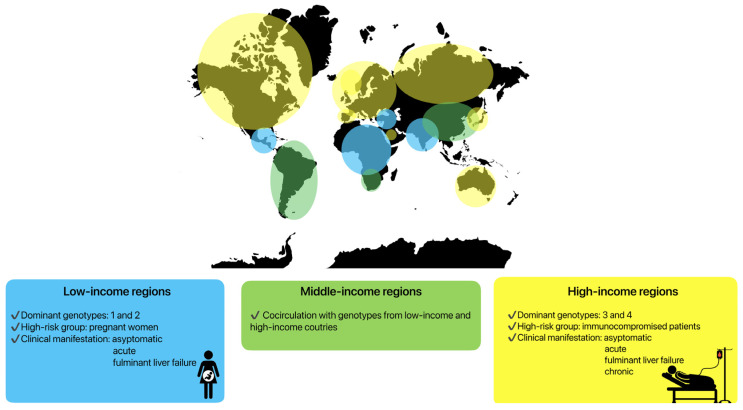
Clinical manifestations of HEV infection depending on specific regions and genotypes.

## Data Availability

No new data were created or analysed in this study.

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
