# Peer review of "Hepatitis E Virus in the Role of an Emerging Food-Borne Pathogen"

_microorganisms, 2025, doi:10.3390/microorganisms13040885_

Round 1
Reviewer 1 Report
Comments and Suggestions for Authors
In this review, the authors present a summary of recent findings related to the epidemiology of Hepatitis E virus (HEV) as an emerging food-borne pathogen. Several revisions on this topic have been published recently, and this manuscript does not introduce any new perspectives or management/mitigation guidelines. Nevertheless, if enhanced, this review could add value to the existing literature. Notably, extensive research has been conducted in various South American countries, as well as a few African nations, yielding intriguing and abundant data that is currently absent from this manuscript. It is essential for the authors to incorporate all these findings in the revised version. Figure 1: It is recommended to include information regarding Genotypes 5, 6, 7, and 8. Figure 2: As a developing region, in Latin America Genotype 3 is the predominant there. I suggest differentiating the data by low/middle-income and high-income regions for more clarity.
Comments on the Quality of English Language
Quality of english language could be improved
Author Response
Comments 1: In this review, the authors present a summary of recent findings related to the epidemiology of Hepatitis E virus (HEV) as an emerging food-borne pathogen. Several revisions on this topic have been published recently, and this manuscript does not introduce any new perspectives or management/mitigation guidelines. Nevertheless, if enhanced, this review could add value to the existing literature.
|
Response 1: We would like to thank you for your feedback. We agree with you, there are several revisions on this topic. However, we still believe this review can bring new point of view. This review has been written in response to increasing number of viral hepatitis E in human population, along with incrising number of studies pointed to widespread circulation of hepatitis E virus (HEV) in animal population. Understanding of HEV have completely changed over the past 20 years. Now it is clear HEV is important ethological agent of viral hepatitis E in low-income regions, as well as in high-income regions (especially for vulnerable groups of people). Given our experiences as veterinarians and scientists working with this zoonotic virus, we identified the necessity to highlight HEV as an emerging food-borne pathogen. In this review we focused on food-borne HEV transmission. We believe our review can bring different point of view, as we have described HEV through all line, from available direct epidemiological links between human cases and consumption of meat or meat products, to presence of HEV in food-producing animals, meat, offal, meat products, milk and shellfish. Our review therefore summarizes current knowledge about HEV circulation in the animals and animal products. It also highlights information that is not available or is still very limited and should be subject of further research. In other sections we focused on HEV epidemiology, clinical manifestation in human population and on preventive measures. We have also described important knowledge gaps and future perspective. We totally believe in the mission of our profession – more attention needs to be given to the presence of HEV among food-producing animals, especially in slaughter time and also to presence of HEV in animal products. We need to prevent food-borne outbreaks and protect vulnerable consumer populations to ensure the health of humans and animals along with food and environmental safety. As we sincerely appreciate and agree with your suggestions, we have added absent knowledge, when they were available.
|
Comments 2: Notably, extensive research has been conducted in various South American countries, as well as a few African nations, yielding intriguing and abundant data that is currently absent from this manuscript. It is essential for the authors to incorporate all these findings in the revised version.
|
Response 2: We would like to thank you for bringing this to our attention. We fully agree with you. As number of viral hepatitis E has risen significantly, especially in Europe countries, we had focused mostly on research in Europe. However, the truth is, that research about the HEV detection has also been realized in various countries apart from Europe, that was absent from this manuscript. We have turned our attention to research in other parts of the world: Africa, Asia, North America, South America and Australia. We have included current knowledge about HEV in different parts of world, wherever it was possible and data were available. (see Section 3, Parts 3.1.1., 3.1.2., 3.1.3.)
Comments 3: Figure 1: It is recommended to include information regarding Genotypes 5, 6, 7, and 8. Response 3: Thank you for your suggestion. We have carefully considered this suggestion and we agree with your opinion. However, Figure 1. was made to declare possible ways of HEV in human population. Considering that up to now genotypes HEV-5, 6 and 8 have been detected only in animal population and the information about their transmission are relatively limited, we are inclined not to add them out of the figure. On the other hand, the transmission of genotype HEV-7 to humans is possible. Based on this fact, the genotype HEV-7 has been added to Figure 1. We fully understand, that figure description was not complete and clear. For a better explanation, we also slightly modified and specified information in figure description. (see Section 3., Figure 1.)
Comments 4: Figure 2: As a developing region, in Latin America Genotype 3 is the predominant there. I suggest differentiating the data by low/middle-income and high-income regions for more clarity. Response 4: Thank you for bringing this to our attention. We have considered this suggestion and we agree with you. Figure 2. has been modified differentiating the data by low/middle-income and high-income regions. (see Section 4., Subsection 4.1., Figure 2.)
|
4. Response to Comments on the Quality of English Language |
Point 1: Quality of English language could be improved. |
Response 1: We hope that quality of English language has been improved. |
|
|

Reviewer 2 Report
Comments and Suggestions for Authors
This manuscript comprehensively summarizes HEV as an emerging food-borne pathogen and it is well written.
Minor comments:
Line 37 should be “Until now, genotypes HEV-5, HEV-6 and HEV-8 have only been detected in animal populations.”
Line 43 should be “The presence of HEV RNA has also been repeatedly published in different products of animal origin.”
Similarly, other sentences should be edited.
HEV antigen and RNA have also been detected in urine. Can author add the point that urine may also pose a risk of HEV transmission?
Can author make a table to summarize 3.1, including information of meat, offal, milk that are HEV positive from which animal species and which genotypes of HEV are detected. Are they confirmed to infect humans? And more information like this. I believe this kind of table will make readers to catch up the points in this part easier.
Author Response
Comments 1: This manuscript comprehensively summarizes HEV as an emerging food-borne pathogen and it is well written. Minor comments: Line 37 should be “Until now, genotypes HEV-5, HEV-6 and HEV-8 have only been detected in animal populations.” |
Response 1: Thank you for your suggestion. The sentence has been corrected. (see Section 1.) |
Comments 2: Line 43 should be “The presence of HEV RNA has also been repeatedly published in different products of animal origin.” |
Response 2: Thank you for pointing this out. The sentence has been corrected. (see Section 1.)
Comments 3: Similarly, other sentences should be edited. Response 3: Thank you for your suggestion. We have checked the entire manuscript and edited text. Comments 4: HEV antigen and RNA have also been detected in urine. Can author add the point that urine may also pose a risk of HEV transmission? Response 4: Thank you for pointing this out. Available information concerning presence of HEV in urine has been added (see Section 3., Part 3.1.1.). Comments 5: Can author make a table to summarize 3.1, including information of meat, offal, milk that are HEV positive from which animal species and which genotypes of HEV are detected. Are they confirmed to infect humans? And more information like this. I believe this kind of table will make readers to catch up the points in this part easier. Response 5: Thank you for bringing this to our attention. We agree with you. The table including information about observed categories, animals, genotypes, as well as confirmation of human infection, has been added as Supplementary Material Table S1. (see Supplementary Material, Table S1)
|
|
|
|

Reviewer 3 Report
Comments and Suggestions for Authors
This nice review explores the emerging issue of HEV infection, analyzing the causes, transmission mechanisms, symptoms, and prevention strategies, with the aim of providing a comprehensive overview of the impact of this pathogen on public health.
I have several detailed suggestions.
Comment 1: Lines 46-47, I would rephrase the sentence this way: “However, the clinical manifestation of infection depends on the HEV genotypes and the patient’s immune system”.
Comment 2: Line 68, There is a missing “a” before “small population”: “Only in a small population of adults…”
Comment 3: Line 102, figure legend 1, There is a missing comma after “water”: “Genotypes 1 and 2 are mainly transmitted by drinking contaminated water, from mother to fetus during pregnancy and by transfusion of infected blood.
Comment 4: Lines 106-167, Subsection 3.1.1 is more general, while section 3.1 analyzes the data in more detail. I would suggest reversing the order of section 3.1 and subsection 3.1.1.
Comment 5: Lines 241-266, Studies that have demonstrated the presence of HEV RNA in milk are listed. I would suggest indicating the genotype of HEV detected in these studies.
Comment 6: Lines 276, I suggest modifying the sentence as follows: “In 2004, hepatitis E in some Japanese patients was connected with…”
Comment 7: Line 362-363: "Infected animals, their meat and offal may contaminate the entire slaughter lines HEV". I suggest modifying the sentence: “Infected animals, along with their meat and offal, can contaminate the entire slaughter line with HEV.”
Comments on the Quality of English LanguageThe manuscript is well written, but Minor editing of English language required
Author Response
This nice review explores the emerging issue of HEV infection, analyzing the causes, transmission mechanisms, symptoms, and prevention strategies, with the aim of providing a comprehensive overview of the impact of this pathogen on public health. I have several detailed suggestions. Comments 1: Lines 46-47, I would rephrase the sentence this way: “However, the clinical manifestation of infection depends on the HEV genotypes and the patient’s immune system”. |
Response 1: Thank you for your suggestion. The sentence has been rephrased. (see Section 1.)
|
Comments 2: Line 68, There is a missing “a” before “small population”: “Only in a small population of adults…” Response 2: Thank you for pointing this out. The missing „a“ before „small population“ has been added to the sentence. (see Section 2.) Comments 3: Line 102, figure legend 1, There is a missing comma after “water”: “Genotypes 1 and 2 are mainly transmitted by drinking contaminated water, from mother to fetus during pregnancy and by transfusion of infected blood. Response 3: Thank you for your suggestion. Missing comma has been added to the sentence. (see Section 3.) Comments 4: Lines 106-167, Subsection 3.1.1 is more general, while section 3.1 analyzes the data in more detail. I would suggest reversing the order of section 3.1 and subsection 3.1.1. Response 4: Thank you for pointing this out. We have carefully considered this suggestion. We have added a short paragraph in section 3.1. to introduce following subsections. We hope that this addition can improve the understanding of the structure of the section and its subsections (see Section 3., Subsection 3.1.). Comments 5: Lines 241-266, Studies that have demonstrated the presence of HEV RNA in milk are listed. I would suggest indicating the genotype of HEV detected in these studies. Response 5: Thank you for your suggestion. Available information of HEV genotypes detected in milk samples have been added to the manuscript. (see Section 3., Part 3.1.4.) Comments 6: Lines 276, I suggest modifying the sentence as follows: “In 2004, hepatitis E in some Japanese patients was connected with…” Response 6: Thank you for pointing this out. The sentence has been modified. (see Section 3., Part 3.1.5.) Comments 7: Line 362-363: "Infected animals, their meat and offal may contaminate the entire slaughter lines HEV". I suggest modifying the sentence: “Infected animals, along with their meat and offal, can contaminate the entire slaughter line with HEV.” Response 7: Thank you for your suggestion. The sentence has been modified. (see Section 5.)
|
|
4. Response to Comments on the Quality of English Language |
Point 1: The manuscript is well written, but Minor editing of English language required |
Response 1: Minor editing of English language has been done. |
|
|

Round 2
Reviewer 1 Report
Comments and Suggestions for Authors
This revised draft represents a significant improvement. While it may not be exhaustive, the review is now prepared for publication. Further enhancements to the English language are necessary.
Comments on the Quality of English LanguageFurther enhancements to the English language are necessary.
Author Response
Comments 1: This revised draft represents a significant improvement. While it may not be exhaustive, the review is now prepared for publication. Further enhancements to the English language are necessary. |
Response 1: |
Thank you for your valuable suggestions. We have considered it and we agree with you. Further enhancements to the English language has been done by Native Speaker Kirstin Macdonald. We hope that this correction leads to the improvement of language quality and complete quality of this review. We sincerely appreciate the time and effort you have invested in evaluating our work.
4. Response to Comments on the Quality of English Language |
Point 1: Further enhancements to the English language are necessary. |
Response 1: Quality of English language has been improved. Correction of English language has been done by Native Speaker Kirstin Macdonald. |
Reviewer 2 Report
Comments and Suggestions for Authors
It looks like there is a problem with the citation in revised manuscript. For example, HEV shedding occurs not only in feces but also in urine, which may contribute to the spread of the virus in the environment and pose a risk of HEV transmission [74,75]. References 74 and 75 look not related to HEV shedding in urine. The author should check the entire manuscript.
Author Response
Comments 1: It looks like there is a problem with the citation in revised manuscript. For example, HEV shedding occurs not only in feces but also in urine, which may contribute to the spread of the virus in the environment and pose a risk of HEV transmission [74,75]. References 74 and 75 look not related to HEV shedding in urine. The author should check the entire manuscript. |
Response 1: Thank you for pointing this out. As we sincerely appreciate your suggestions, we have carefully checked the entire manuscript and we have done minor changes. However, references 74 and 75 are related to HEV shedding in urine. Reference 74 (Banks et al., 2004) has confirmed the occurrence of HEV also in urine of infected pigs. Reference 75 (Bouwknegt et al., 2009) has confirmed the occurrence of HEV also in urine of infected pigs and urine was identified as possible HEV source for pig-to-pig and pig-to-human HEV transmission. We hope the references in the entire manuscript are now correct. We sincerely appreciate the time and effort you have invested in evaluating our work. |
Reviewer 3 Report
Comments and Suggestions for Authors
I appreciate the work done by the authors and that they accepted my suggestions. In my opinion, the paper is now ready for publication.
Author Response
Comments 1: I appreciate the work done by the authors and that they accepted my suggestions. In my opinion, the paper is now ready for publication. |
Response 1: We would like to thank you for your feedback. We sincerely appreciate the time and effort you have invested in evaluating our work.
|